# Characterization of Nervous Necrosis Virus (NNV) Nonstructural Protein B2 and Its Enhancement on Virus Proliferation

**DOI:** 10.3390/v14122818

**Published:** 2022-12-17

**Authors:** Yuqi Zhang, Fujing Dong, Jing Xing, Xiaoqian Tang, Xiuzhen Sheng, Heng Chi, Wenbin Zhan

**Affiliations:** 1Laboratory of Pathology and Immunology of Aquatic Animals, KLMME, Fisheries College, Ocean University of China, Qingdao 266003, China; 2Laboratory for Marine Fisheries Science and Food Production Processes, Qingdao National Laboratory for Marine Science and Technology, Qingdao 266071, China

**Keywords:** nervous necrosis virus, B2 protein, RNA3, NNV replication, overexpression

## Abstract

The nerve necrosis virus (NNV), a pathogen of viral nervous necrosis disease in several important mariculture economic fish species, causes economic loss. Its nonstructural protein B2 encoded by the sub-genomic RNA3 affects the amplification of the virus. In this study, the B2 protein was recombinantly expressed, the polyclonal antibodies were produced and the dynamics of the B2 protein and genomes were measured in vivo and in vitro after NNV infection. Then, the effects of the overexpressed B2 protein on virus proliferation were investigated. The results showed that the polyclonal antibodies can recognize the B2 protein in both SSN-1 cells and the brain/eye of the grouper. The RNA3 expression significantly increased at 12 h and kept rising till the end of the experiment; it was 10^6.9^ copies/μL at 120 h. The B2 protein could be first detected at 3 h post-infection, which was earlier than the capsid protein was first detected (12 h post-infection). The B2 protein can be detected in the brain, eye and heart on day 3 and the copy number of genomes reached a maximum at 6 d post-infection. There was a low expression of NNV genomes in the liver, spleen and kidney, and no virus was detected in the gill, stomach and intestine. In the meantime, the B2 protein was successfully expressed in GF-1 cells and significantly enhanced virus proliferation, which produced an earlier cytopathic effect and higher cell death rates after 3 d post-infection than the control. In conclusion, the B2 protein acts as an early expressed protein during virus replication and proliferation and is involved in the early infection of NNV. The results may provide insight into the early stage of virus infection and prevention of the disease.

## 1. Introduction

Viral nervous necrosis (VNN), which also includes viral encephalopathy and retinopathy (VER), is one of the most causative and infectious diseases affecting both cultured marine and freshwater species of fish worldwide [1]. VNN-infected fish show darkening, exophthalmos, abnormal swimming behavior and vacuolation and necrosis of nervous tissues (brain and retina) [2]. VNN normally targets the larval and juvenile stages of fish, while mortalities in adult fish were also reported [3,4].

The pathogen of VNN is the nervous necrosis virus (NNV), which belongs to Nodaviridae, *Betanodavirus*. The Nodaviridae family contains two genera: *Alphanodavirus* and *Betanodavirus*, infecting insects and fish, respectively [5]. The particle of NNV is spherical or icosahedral without an envelope in a diameter of about 25–35 nm. Its genome consists of two single-stranded positive-sense RNAs named RNA1 and RNA2 [6]. RNA1 (3.1 kb) encodes RNA-dependent RNA polymerase (RdRp), and RNA2 (1.4 kb) encodes the only structural protein of the virus capsid protein [7,8]. RNA3, a sub-genomic RNA species from the 3′ termini of RNA1, is synthesized in the progress of an infection. RNA3 encodes two early express nonstructural viral proteins: B1 (~11 kDa) and B2 (~8 kDa) [9]. The B1 protein consists of 336 nucleotides and its open reading frame (ORF) matches the 3′ termini of the RdRp reading frame. Several studies indicated the function of the B1 protein as an anti-necrotic death factor to guarantee the implication of the virus, which decreases the loss of mitochondrial membrane potential (MMP) and up-regulates p53/p21 to induce G1/S cell cycle arrest [10,11,12]. The B2 protein consists of 228 nucleotides and its ORF is in a + 1 reading frame relative to RdRp. In contrast, the B2 protein plays an essential role in intracellular necrotic infection.

The B2 protein binds to double-stranded DNA (dsDNA) to prevent the cleavage of Dicer, while antagonizing RNA interference (RNAi) and RNA silence from the host anti-virus response, accumulating the implication of RNA1 [13,14]. Through the targeting signal peptide, the B2 protein increases the loss of MMP to induce the ATP-deletion death of cells [15]. On this basis, Su demonstrated that the B2 protein up-regulates the expression of the apoptotic gene Bax, causing the loss of MMP as well. Additionally, the B2 protein inhibits the type I interferon (IFN) response by suppressing host transcription directed by RNA polymerase II [16,17].

Several pieces of research have explored the detection of NNV genomes, including quantitative reverse transcriptase PCR (qRT-PCR), in situ hybridization (ISH), and immunohistochemistry (IHC) [18,19,20]. Kim optimized ISH by utilizing the two-color labeling RNA–RNA probes. Compared to the traditional technique, molecular detection methods have more efficiency, sensitivity and accuracy [19]. However, the reports listed above concentrate on the distribution or viral replication of genomes. Only a few reports explored the function or characterization of the nonstructural protein B2, as it plays various roles in the early infection of the virus.

The aim of this study was to explore the expression characteristics of the B2 protein during viral infection in vivo and in vitro by producing a specific antibody and this study further explored the function of the B2 protein on the proliferation of the virus, which advances the death of cells. The B2 gene and RNA3 gene sequences were predicted from the gene sequences of the RGNNV (CNPgg2108 strain) virus strain published in NCBI (RNA1 sequence number: MT157513; RNA2 sequence number: MT157514) and their respective specific primers were designed to express the B2 recombinant protein and prepare B2-specific antibodies. The qRT-PCR technique was used to analyze the viral genomic changes during virus infection of SSN-1 cells and grouper larvae; the IIFA technique analyzed the dynamic study of B2 protein co-expression with the capsid protein during virus infection of cells; the IHC technique analyzed the distribution of the B2 protein in neural tissues and the influence of B2 protein overexpression in GF-1 cells was detected.

## 2. Materials and Methods

### 2.1. Virus and Cell Culture

NNV (CNPgg 2108 strain, RGNNV genotype) [21] was propagated in striped snakehead cells (SSN-1) and grouper fin cells (GF-1) (SSN-1 was kindly bestowed by the Shenzhen Entry–Exit Inspection and Quarantine, GF-1 was kindly provided by Professor Yinghui Qin) [17]. The SSN-1 cells and GF-1 cells were grown in 25 cm^2^ tissue culture flasks at 25 °C cultured in Leibovitz’s L-15 medium (Gibco, Grand Island, NY, USA) containing 10% FBS, 100 IU/mL penicillin (Gibco, USA) and 100 µg/mL streptomycin (Gibco, USA).

Virus was inoculated onto cells and the cell cultures were incubated at 25 °C until extensive complete cytopathic effect (CPE) was observed. Supernatants were harvested and centrifuged (12,000× *g*, 10 min 4 °C) to remove cell debris. SSN-1 cell lines were used for determining viral titers. The 50% tissue culture infection dose (TCID_50_) was determined using the methodology described by Reed and Muench [22]. The virus applied in the study for viral infection was in coarse extraction mixed with L-15 medium (2% FBS and 1% Penicillin–Streptomycin) and detected by the electron micrograph in our previous study [21]. After SSN-1 cells had floated and died entirely, supernatants mixed with cell debris were harvested and freeze–thawing was repeated 3 times to adequately release the virus. The mixture was centrifuged (12,000× *g*, 10 min 4 °C) and filtrated with 0.22 μm (Merck Millipore, Burlington, MA, USA).

### 2.2. Antibodies and Animals

The monoclonal antibody (3A6) of red-spotted grouper nervous necrosis virus capsid protein (Cp) was produced previously in our laboratory [23] and used in indirect immunofluorescence assay (IIFA).

The genes of B2 protein and Cp were amplified using specific primers listed in Table 1 and in our previous study [23], respectively. The PCR product was purified and inserted into pET-32a prokaryotic expression vector. The recombinant proteins were affinity-purified using His Trap HP Ni-Agarose (GE healthcare, China) according to the manufacturer’s instructions [24]. The concentrations of purified proteins were determined by the Bradford method. The polyclonal antibodies of rabbit anti-red-spotted grouper nervous necrosis virus B2 protein (anti-B2 protein) and rabbit anti-red-spotted grouper nervous necrosis virus Cp (anti-Cp) were produced as described previously [25]. The specificity of Abs was analyzed using Western blot and IIFA.

Healthy New Zealand white rabbits weighing 1.5 kg were purchased from Qingdao Drug Inspection Institute (Shandong, China) and kept in the laboratory for two weeks, and then were used for antibody production.

The larvae pearl gentian groupers with an average weight of 7.32 g and a body length of 5–7 cm were purchased from a farm (Rizhao, Shandong, China), and were temporarily reared in a tank with a water temperature of 27 °C–28 °C for two weeks before the experiment began, with regular baiting and seawater replacement and continuous oxygenation.

### 2.3. In Vitro NNV Infection

SSN-1 cells were placed on 24-well cell culture slides and incubated at 25 °C until growth confluence. Multiplicity of infection (MOI) for NNV was 1.0. SSN-1 cells were infected with virus at a titer of 10^5^ TCID_50_/mL, and cells were collected from both culture slides at 0 h, 3 h, 6 h, 12 h, 24 h, 36 h, 48 h, 72 h, and 120 h post-infection and NNV was detected and localized. The supernatant was removed from each cell culture slide and the adherent cells were fixed with 4% paraformaldehyde for 15 min for indirect immunofluorescence analysis. The supernatant was taken from the remaining slides and the adherent cells were washed once with diethylpyrocarbonate (DEPC)-treated water for qRT-PCR of RNA1, RNA2 and RNA3 in a closed water system.

### 2.4. In Vivo NNV Challenge

Healthy fish were selected for the experiments. The 100 fish were divided into two groups: control (*n* = 50) and sampling group (*n* = 50). According to the ID_50_ in pre-experiments, the fish in the sampling group were injected intramuscularly with 100 μL 10^5^ ID_50_/mL of NNV for the follow-up experiments. The fish in the control group were injected intramuscularly with 100 μL of Leibovitz L-15 culture solution. Five fish were randomly collected at 3, 6, 12, 24, 48 days post-inoculation from the control non-infected group and from the challenged group. Animals collected from challenged group were alive, either apparently healthy or with typical signs of the disease and the following organs/tissues were sampled: brain, eye, heart, liver, spleen, kidney, gill, stomach, intestine, and muscle.

Organs from three fish were pooled together at each sampling point, immediately frozen in liquid nitrogen, and stored at -80 °C until used. Three of these pools were used for the quantification of NNV genomes by qRT-PCR. The virus presence was also confirmed by the subsequent virus genome detection with a specific RT-PCR [18]. For the Immunohistochemical analysis, the fish from the infected and non-infected groups were sampled at the times indicated above. Brains and eyes were collected and fixed in buffered formalin at room temperature for 24 h. Excess fixative was removed by washing with DEPC water for 1 h and the organs were maintained in 70% ethanol at room temperature until used.

### 2.5. Transient Transfection and Virus Infection

The recombinant plasmids of pcDNA3.1-B2 were produced as per Section 2.2 employing the plasmid of pcDNA-3.1. The transfection with Lipofectamine 3000 Reagent Protocol (Invitrogen, Waltham, MA, USA) was according to the manufacturer’s instructions. Meanwhile, the cells transfected with pcDNA3.1 and the un-transfected cells acted as the control group. Plasmids were mixed with Lipofectamine 3000 in Opti-MEM (Gibco, USA) for 15 min and added to GF-1 cells seeded in 6-well plate incubating for 6 h. After that, the mixture was aspirated and washed with PBS and then cultured with L-15 medium (2% FBS and 1% Penicillin–Streptomycin) for 24 h. The transfection efficiency was detected by qRT-PCR. The total RNA was extracted and the mRNA levels of B2 protein were examined by qRT-PCR primers of RNA3 (Table 1), which contain the main domain of B2 protein. The β-actin gene (Table 1) was used as an internal control and the relative expression was represented as fold induction relative to the expression level in control cells. The Ct values of samples more than 35 were calculated as 35.

For virus infection assays, GF-1 cells were infected with NNV (MOI = 1) at 25 ℃. After 2 h of virus absorption, cell monolayers were washed with PBS and subsequently maintained in L-15 medium (2% FBS and 1% Penicillin–Streptomycin). After 48 h of infection, cells were harvested and washed twice with PBS. Later, cells were lysed with RIPA buffer (Beyotime Biotechnology, China) and Western blot was performed to detect the expression of NNV Cp. Another batch of cells at different infection time points was detected for cell viability.

### 2.6. RNA Extraction and Reverse Transcription

The viral RNA from SSN-1 cells and GF-1 cells was extracted using a RNeasy Mini Kit (BioMarker) according to the manufacturer’s protocols. The organs/tissue total RNA was extracted by TRIzol method according to the manufacturer’s instructions and the RNA was treated with the DNase to remove trace amounts of DNA after elution from the filter. The quantity and concentration of the RNA were detected by NanoDrop 8000 spectrophotometer (Thermo Scientific, Waltham, MA, USA) and the cDNA was synthesized by using Reverse Transcriptase M-MLV kit (TaKaRa, Beijing, China) according to manufacturer’s instructions.

### 2.7. Generation of Standard Curves for RNA Absolute Quantification

The absolute qRT-PCR method was described previously [26]. Standard curves were obtained using 10-fold serial dilutions of in vitro transcribed RNA. RT-PCR products resulting from the amplification of the RNA1 and RNA3 segments were purified from agarose gels. The primer information is in Table 1. The target fragments were recovered by agarose gel electrophoresis and ligated to the T vector according to the instructions. The RNA1 and RNA3 genomic plasmids were extracted using the Plasmid Mini Extraction Kit and checked by Nanodrop 8000 ultra-micro spectrophotometer. The plasmid standards were diluted with DEPC water in a 10-fold gradient and used as templates for fluorescent quantitative PCR amplification. Three parallel experiments were set up for each group. After the reaction, the parameters obtained such as the lysis curve and amplification curve were analyzed and the logarithmic value of the template copy number was used as the horizontal coordinate and the Ct value of each plasmid standard as the vertical coordinate, and the standard curve was plotted using the software. The standard curve of RNA2 was generated in our previous study [23].

### 2.8. SYBR Green Quantitative Real-Time PCR Assay In Vitro and Vivo

The synthesized cDNA from SSN-1 cells and organs/tissue described above was diluted for the qRT-PCR. The expression levels of NNV genomes in vivo and in vitro were determined with the LightCycler 480 PCR system (Roche) using the SYBR Green Realtime PCR Master Mix (TaKaRa, China). β-actin primers (β-actin F/β-actin R) were introduced as the reference genes (Table 1). The cycling protocol was: 95 °C, 30 s; (95 °C, 15 s and 60 °C, 5 s) × 40 cycles; 95 °C, 5 s; 65 °C, 60 s. Melting curve analysis and sequencing were used to detect the specificity of PCR products. All samples were analyzed in triplicate. Significant differences between the number of copies of each viral genome segment at different times were calculated by one-way ANOVA, followed by Fisher’s least significant difference (LSD) test. Statistical analysis was performed using GraphPad Prism 6.0 software. 

### 2.9. Indirect Immunofluorescence Assay (IIFA) 

The SSN-1 cell crawls were washed in PBST for 5 min after fixation with 4% paraformaldehyde. The cells were subsequently blocked with 5% BSA for 1 h at 37 °C, the antibody was incubated for 1 h at 37 °C in a constant temperature chamber, and the primary antibody was carefully removed by aspirating the gun tip and washed twice in PBST for 10 min each time. The mixed antibody of Alexa Fluor 488-labeled sheep anti-mouse IgG antibody and Alexa Fluor 649-labeled sheep anti-rabbit IgG antibody was diluted 2000 times as secondary antibody and incubated for 45 min at 37 °C. To reduce the loss of cells during elution, DAPI diluted 3000 times was added to the secondary antibody at the same time for nuclear re-staining and washed twice for 8 min each time in PBST. Fluorescence images of the samples were obtained using fluorescence microscope (Olympus, Japan). 

### 2.10. Immunohistochemistry (IHC) 

Brain and eye tissues from uninfected and infected fish were placed in Boone’s fixative and then soaked in freshly prepared 75% ethanol for 12 h, 24 h and 48 h. The tissue was then sliced into 7 µm-sections and dried overnight at 37 °C. The fixed samples were dehydrated in 70%, 80%, 95% and 100% alcohol for 30 min, with 95% and 100% alcohol being dehydrated twice each, and then the tissues were placed in a 1:1 mixture of xylene and alcohol and pure xylene for 30 min. The tissue was then placed in a 1:1 mixture of xylene and alcohol and pure xylene for 30 min, after which it was placed in a mixture of xylene and paraffin wax for 30 min. The sample sections were dewaxed twice in xylene for 10 min each, and then in alcohol at 100%, 95%, 80%, 70% and 50% for 5 min each, and soaked in PBS for 5 min. The slides were then immersed in citrate buffer solution and placed in a water bath at 60 °C for 15 min and soaked in PBS for 3 min for antigen repair.

The samples were incubated for 30 min in a 37 °C thermostat with a drop of BSA blocking solution, and the excess liquid was shaken off with a flaker without washing. Rabbit anti-B2 protein polyclonal antibody diluted as primary antibody was incubated for 90 min in a 37 °C thermostat or overnight in a 4 °C refrigerator and washed three times with PBST for 5 min each. When the red signal was visible and the color was appropriate, the coloring reaction was terminated by washing with ultrapure water; the hematoxylin was re-stained for about 5 min and the excess dye was washed off with pure water after re-staining and the adherent slides were rinsed with pure water again after a few seconds of alcohol fractionation in hydrochloric acid. The slides were then dipped sequentially into 50−100% drops of alcohol and sealed with a neutral gum solution after being transparent in xylene. The staining results were observed under a light microscope (Olympus, Tokyo, Japan) and photographed for documentation. The intensity of positive signal was quantified by Image J 1.51k.

### 2.11. Western Blot Assay

Western blot was performed as described previously [23]. The extracted protein from un-transfected (negative) and transfected cells (pcDNA3.1 and pcDNA3.1-B2) was separately mixed with an equal volume of 2× protein loading buffer and denatured for 5 min in boiling water. Subsequently, the prepared samples were resolved in SDS-PAGE and electrophoretically transferred onto polyvinylidene fluoride (PVDF) membranes (Merck Millipore, Darmstadt, Germany) using transfer buffer (25 mM Tris, 192 mM Gly, 3.5 mM SDS). After blocking with 5% BSA for 1.5 h at 37 °C, membranes were probed with different primary antibodies for 1 h at 37 °C. Following three washes with PBST, horseradish peroxidase (HRP)-conjugated goat-anti-rabbit IgG (Jackson, USA) at a 1:20,000 dilution was added and further incubated for 45 min. After rinsing as above, blots were developed with an enhanced chemiluminescent kit (Vazyme, Nanjing, China) for 1 min in the dark, and then visualized with a chemiluminescence imaging system (FUSION FX Spectra, Vilber Lourmat, Collégien, French). In the transfection assay, the expression of GAPDH was used as the internal control.

### 2.12. Cell Viability Analysis

Cell viability of GF-1 cells in 96-well plate was assessed with cell counting kit-8 (CCK-8, Vazyme, China) depending on the instruction. Cell survival fraction (%) = [(OD450 of treated cells − OD450 of L-15)/ (OD450 of untreated cells − OD450 of L-15)] × 100%. Untreated cells represented GF-1 cells without transfection and infection at all time points. The CPE of GF-1 cells in 6-well plate was visualized with inverted microscope (Olympus, Japan).

### 2.13. Statistics Analyses

The statistical analysis was implemented utilizing GraphPad Prism 6.0 software and all data were shown as mean ± standard deviation (SD). The differences between the data were analyzed by one-way analysis of variance (ANOVA). A *p*-value < 0.05 (*p* < 0.05) was considered statistically significant. A *p*-value < 0.01 (*p* < 0.01) was considered highly significant.

## 3. Results

### 3.1. The Characterization of B2 Protein Antibody

SDS-PAGE experiments were carried out using the total protein of the B2-expressing bacteria and purified B2 recombinant protein as substrates, respectively. The results showed that there was an obvious thickened band at 29 kDa after IPTG induction and its size was consistent with the theoretical value, while the strain without IPTG induction did not have a thickened band, indicating that the B2 recombinant protein was successfully induced and expressed. The purified B2 recombinant protein had a single target band at 29 kDa (Figure 1A) and the Western blot results showed that the prepared antibody specifically recognized the B2 recombinant protein and emitted a band at 29 kDa, which was consistent with the theoretical size of the B2 recombinant prokaryotic expression protein, while the control group of negative rabbit sera did not emit a band (Figure 1B) indicating the good specificity of the prepared antibody.

The results of indirect immunofluorescence showed that positive fluorescent signals were detected in the rabbit anti-B2 protein polyclonal antibody incubated drops, while no positive fluorescent signals were detected in the control rabbit serum drops of the virus-infected cells (Figure 2). The rabbit anti-B2 protein polyclonal antibodies were able to recognize the natural B2 protein in SSN-1 cells and had no cross-reaction with the Cp monoclonal antibody. The specificity and stability of the antibodies are good and can be used for subsequent studies on the expression characteristics of the B2 protein during in vitro and in vivo virus infection.

### 3.2. Specificity and Sensitivity of the Real-Time PCR Protocols

The real-time PCR protocols developed were also evaluated by the analysis of melting curves; Tm of the RNA1 amplicon is 84 °C and the final standard curve equation obtained for the RNA1 genome was y = −3.84x + 45.382 with a correlation coefficient R^2^ = 0.9994 (Figure 3B). The Tm of the RNA3 amplicon is 90 °C and the final standard curve equation obtained for the RNA3 genome was y = −3.6236x + 40.744 with a correlation coefficient R^2^ = 0.99894 (Figure 3D). This shows that the constructed RNA1 and RNA3 genomic fluorescent quantitative standard curves have a good linear relationship.

### 3.3. The Dynamics of Genome Expression in NNV-Infected SSN-1 Cells 

At 24 h after infection with NNV, the surface lines of the cell line were wrinkled and cellular aggregates were seen (Figure 4B). At 48 h after infection, vacuoles began to appear in the cells with the accumulation of dead cells (Figure 4C). Until infection at 12 h, the copy number of RNA1 was less than 10^6.1^ copies/μL, RNA2 remained at a low level of fewer than 10^3.6^ copies/μL and RNA3 less than 10^5.3^ copies/μL. They all kept rising till the end of the experiment from 24 h to 120 h, with RNA1 at 10^8.2^ copies/μL, RNA2 at 10^5.6^ copies/μL and RNA3 at 10^6.9^ copies/μL (Figure 4F). RNA2 gene expression was lower than RNA1 and RNA3 expression at all time points, indicating that RNA1 and RNA3 fragments were involved in the early stages of viral replication and that RNA2 expression induced cell necrosis, with increasing viral particles after 12 h.

### 3.4. The Co-Expression of B2 Protein and Cp

The co-expression dynamics of the Cp and B2 protein after viral infection of SSN-1 cells were studied using indirect immunofluorescence. A green fluorescent signal represents natural B2 protein, a red fluorescent signal represents Cp, and a blue fluorescence is DAPI re-stained nuclei (Figure 5). The results showed that after NNV infected SSN-1 cells, the B2 protein started to be expressed after 3 h. Before 12 h, there was no detection of Cp. At 36 h, the green fluorescent signals of the B2 protein intensified and started to accumulate from the cytoplasm to the nucleus, and some cells showed co-expression of Cp. Cells started to show symptoms of lysis and death, and both red and green fluorescent signals accumulated at the nucleus after 72 h. It is shown that the B2 protein is an early expressed protein in the process of virus proliferation, preparing the virus particles for proliferation and that the proliferation of the capsid protein after 12 h drives the cells to lysis, causing continuous cell rupture and death.

### 3.5. The Tissue Distribution of Genome in Experimentally Infected Larvae Grouper

As is shown in Figure 6A, negatively stained intact or empty viral particles existed in the virus from infected-SSN-1 cell culture fluid which injected grouper larvae. RT-PCR qualitative analysis of the B2 protein gene expression content in tissue and organs of larvae grouper at different time points after NNV infection showed that the B2 protein had a high expression level in the brain, eyes and heart and it was not detected in the gill, stomach, intestine and muscle tissues at any other time points of infection, except for trace B2 protein gene expression in muscles sampled 3 days after infection (Figure 6B).

The expression of RNA1, RNA2 and RNA3 in the tissues of infected larvae fish was measured by fluorescence qRT-PCR at different time points. RNA1, RNA2 and RNA3 copy numbers all reached their highest values on day 6 in the brain, eye and heart tissues (Figure 6C). The maximum of RNA1 and RNA3 copy numbers was in the eye with 10^7.9^ copies/μL and 10^7.1^ copies/μL, respectively. RNA2 copy numbers reached the highest with 10^5.2^ copies/μL in the brain. It was observed that the genome expression levels in the heart were as high as in the eye and brain, which was novel to explore in the future. In other tissues (kidney, liver and spleen), the expression of RNA1 and RNA3 did not exceed 10^4.3^ copies/μL, and the RNA2 copy number was no more than 10^2.3^ copies/μL.

### 3.6. The Detection of B2 Protein in Target Issues by Immunohistochemical Analysis

Immunohistochemistry was applied to detect the expression level of the B2 protein in the neural tissues of NNV-infected grouper larvae. The results showed that the red positive signal was evident in the inner retinal choroidal cells and weaker in the brain gray matter, and the B2 protein was detected in the cytoplasm of both ocular and ocular cells, while no red positive signal was shown in the healthy neural tissues of the grouper larvae (Figure 7).

### 3.7. Overexpression of B2 Protein in GF-1 Cells

To investigate the function of the nonstructural B2 protein on viral replication and cell survival, the transfection results of the eukaryotic plasmid pcDNA3.1-B2 in GF-1 cells were explored. Meanwhile, the empty plasmid pcDNA3.1 and the un-transfected cells (negative) were designed as the control groups. The Western blot results for detecting Cp showed that the B2 protein has a significant enhancement in virus proliferation (Figure 8B), which may be the reason for the extensive death of cells. In Figure 8C, cell viability results illustrated that after infection for 3 d, the survival rates of the pcDNA3.1-B2 transfected group decreased to 55%, showing significant differences with the negative group (69%) and pcDNA3.1-transfected group (65%). The same difference displayed on 5 d. At 7 d, the survival rate was not significant between the transfection groups. Since the influence of transfection on GF-1 cells, the cell number of transfection groups was lower than the negative group at all times. The CPE results indicated that the B2 protein caused extensive and advanced cell death (Figure 8D). Compared with other groups, GF-1 cells in the transfection of pcDNA3.1-B2 displayed a substantial number of cells floating and swelling with large bubbles from the plasma membrane from day 3 to day 7.

## 4. Discussion

NNV causes retinal and neurological vacuolation necrosis in marine and freshwater fish worldwide [27] and is one of the very few minimal animal viruses in the world. The B2 protein plays an important role in viral proliferation and was shown to bind to dsRNA after targeted mutagenesis and to cause cell death through the degradation of mitochondrial complex II [16,28]. In this section, specific primers were designed based on the predicted B2 gene of strain NNV-CNPgg2108, which was amplified by PCR and the results of its agarose gel electrophoresis matched the theoretical nucleic acid size. A B2 recombinant protein of high purity was recombinantly expressed to produce a rabbit anti-B2 protein polyclonal antibody. Because the B2 protein itself has a molecular weight of only 8.5 kDa and the B2 gene is expressed only during the beginning of viral proliferation, it is difficult to obtain the B2 natural protein by collecting the protein from lysed cells through viral infection. The rabbit anti-B2 protein polyclonal antibody prepared in this experiment can be used to verify the presence of the NNV B2 protein at the protein level by indirect immunofluorescence techniques, and can also be used as a validated and functional detection tool for subsequent localization in virus-infected SSN-1 cells and larvae fish tissue.

Real-time quantitative PCR is widely used in the detection of many viruses because of its accuracy, specificity and sensitivity, which overcomes the tendency to produce false-positive results during normal PCR amplification [29]. The NNV genome consists of two single-stranded positive RNA, RNA1 and RNA2. A sub-genomic RNA from the 3′ end of the RNA1 genome called RNA3 is transcribed during viral proliferation and replication. In this chapter, we present the gene sequence of the RGNNV (strain CNPgg2108) strain according to Xing’s publication in NCBI (RNA1 sequence number: MT157513; RNA2 sequence number: MT157514) and predicted the RNA3 gene sequence and designed the respective specific primers. We applied the SYBR green dye method to construct fluorescent quantitative PCR standard curves for the RNA1 and RNA3 genomes. It is mentioned that the standard curve for the RNA2 genome was constructed in our laboratory before, and the amplification equation was: y = −3.4436x + 36.76, with a correlation coefficient of R^2^ = 0.9983, and the melting curve of the amplified product had only one peak (85 °C) [23].

In the fluorescence quantitative PCR experiments of the virus-infected SSN-1 cell line, it was observed that the expression trends of RNA1 and RNA3 at different infection time points were consistent, indicating that RNA3 is a sub-genomic segment transcribed from the nested 3′ end of the RNA1 genome. In addition, the copy numbers of the RNA1 and RNA3 genomes were significantly higher than those of the RNA2 genome in virus-infected cells at different time points, which could be explained by the fact that RNA1 encodes RNA polymerase, which has an important role in viral RNA production, assembly and activation. Venter et al. found that in α-nodavirus, after the synthesis of RdRp reached a certain level in the early stages of viral replication, the translation of RNA2 was up-regulated. While RNA3 was involved in the replication of RNA2, which acts as a trans-activator of RNA2 and started to be down-regulated at the time of RNA2 synthesis. In Kim’s study of β-nodavirus replication, RNA1 was found to be expressed early in viral replication, RNA2 was expressed late, and the overexpression of RNA1 and RNA3 in the early period could ensure the establishment of the early viral replication complex [20,30].

It was found that the sub-genomic RNA3 was responsible for encoding two nonstructural proteins, B1 and B2 [31,32] and that the open reading frame of the B1 protein is identical to that of the RNA polymerase. The B1 protein is an early expression gene that contributes to the anti-necrotic death function during the early replication phase of the virus and can regulate cell death by reducing the loss of mitochondrial membrane potential [10]. The B2 protein, which is encoded by the open reading frame shift + 1 of the RdRp, consists of 75 amino acids and has a relative molecular weight of approximately 8 kDa, and is also expressed early in viral infection. The α protein is a precursor of the RGNNV capsid protein and was found to cause loss of mitochondrial permeability in the early stages of viral infection and trigger intracellular death signals and induce apoptosis at later stages [33,34]. In this study, qRT-PCR experiments revealed that RNA2 expression was low until 12 h of virus infection in SSN-1 cells and gradually increased after 12 h of infection. The increasing level of RNA2 expression after 12 h of infection may be related to the assembly of viral particles, suggesting that RNA2 expression induces apoptosis or necrosis. The B2 protein, as an early expressed protein, may facilitate the assembly of the capsid protein. After 12 h of infection, the virus starts to lyse cells in preparation for infection with a new host and therefore the capsid protein is expressed in large amounts at both the gene and cellular levels during the assembly phase.

In this paper, the distribution of the NNV genome in the tissues of larvae grouper and the distribution of the B2 protein in the neural tissues were investigated by qRT-PCR and IHC, respectively. qRT-PCR showed that high expression of the genome was detected in the eye, brain and heart tissues throughout the virus infection, while the non-neural tissues of the liver, spleen and kidney showed low expression. However, in the RT-PCR experiments, weak viral gene expression was detected in muscle tissues 3 days after infection. No viral genome expression was detected in the gill, stomach, intestine and muscle tissue thereafter, suggesting that the virus may have spread to neural tissue in the brain at the site of inoculation and to other parts of the brain via the blood system. The transfer of the NNV virus from the brain to the eye via the optic nerve was demonstrated as the presence of nodavirus in the blood of several fish species [35,36,37]. Lopez-Jimena found that viral RNA1 and RNA2 fragments were detected in both neural and non-neural tissues at all the time points sampled. There was no increase in viral titers detected in non-neural tissues, suggesting that the maturation of viral particles after these tissues prevented the completion of the viral replication cycle [18]. The highest expression levels of RNA1 and RNA2 in the Lopez-Jimena and Dalla studies started on day 10 and 7, respectively, while the highest expression levels of the RNA1, RNA2 and RNA3 genomes in this experiment started on day 6 and the values of RNA2 copies obtained were all different. This difference may be due to factors such as virus strain, virus dose, water temperature and the condition of the fish used in the experiment [18,38]. The RNA1 and RNA3 genome copy numbers were significantly higher than the RNA2 genome in different tissues at each sampling time point, and the establishment of RNA1 and RNA3 gene fragments is important for detecting the early stages of virus onset. 

In the experiments, the eye and brain nerve tissues with the highest viral genome expression were selected and the results of immunohistochemistry showed that the B2 protein was detectable in the brain and eye 3 d after virus infection, and no capsid protein expression was detected. By the overexpression of the B2 protein in GF-1 cells, B2 acted as the early expression protein, enhancing the replication of the virus. However, due to the slight changes to cells in the 6-well plate without significance, there were no statistics of genomes presented by qRT-PCR, regrettably. Instead, we detected the cell viability and CPE to prove the effect of the B2 protein on host survival. Western blot showed that the B2 protein as a nonstructural protein did have an evident enhancement on virus proliferation, as the fish cell line had lower transfection efficiency.

Both in vitro and in vivo viral infection experiments suggest that the B2 protein as an early expressed protein may play an important role in viral replication and proliferation and particle assembly. The results may provide insight into the early stage of virus infection and the prevention of the disease.

## Figures and Tables

**Figure 1 viruses-14-02818-f001:**
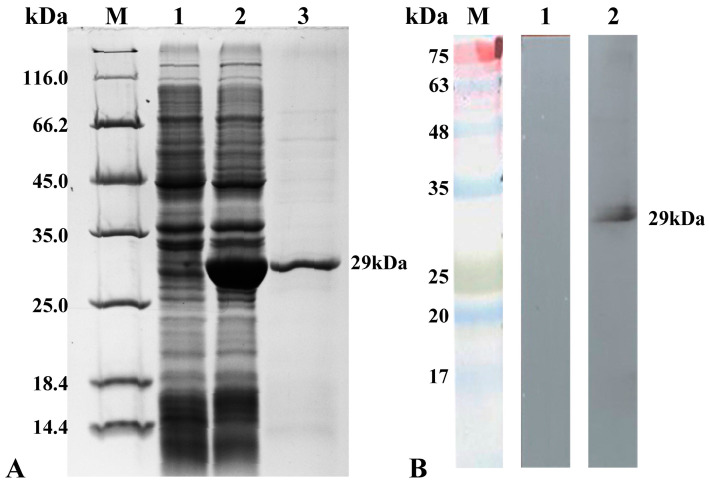
SDS-PAGE and Western blot results of B2 Abs. (**A**) SDS-PAGE results of B2 recombinant protein induction and purification. 1: Total protein of positively expressing bacteria without IPTG induction; 2: Total protein of positively expressing bacteria with IPTG induction; 3: Purified B2 recombinant protein. (**B**) The results of antibodies to recombinant proteins B2; 1: negative control; 2: rabbit anti-B2 protein antibody.

**Figure 2 viruses-14-02818-f002:**
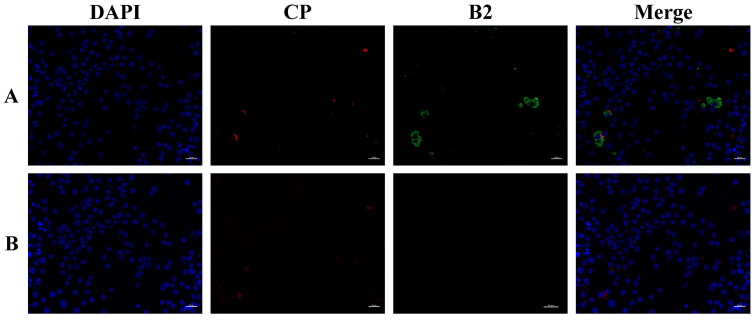
Indirect immunofluorescence results of antibody localization to B2 natural protein in SSN-1 cells infected with NNV. (**A**) Incubation antibody is rabbit anti-B2 protein polyclonal antibody with monoclonal antibody of NNV Cp; (**B**) rabbit negative serum control. Scale bar = 20 μm.

**Figure 3 viruses-14-02818-f003:**
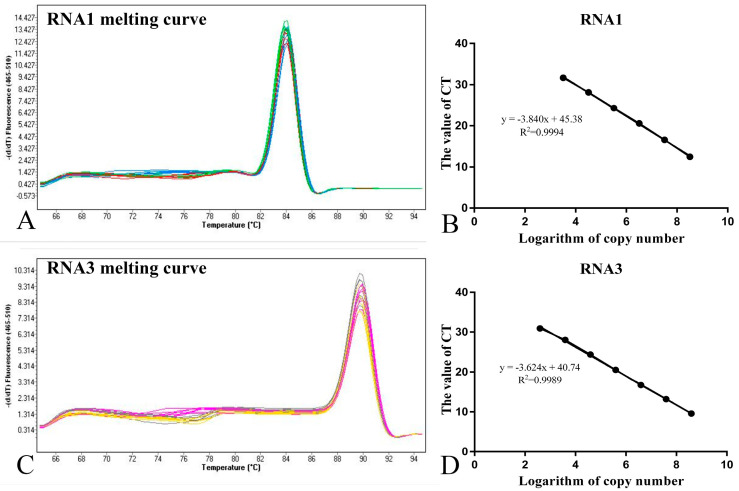
Melting temperature (Tm) and standard curve analyses after amplification of specific RNA1 and RNA3 templates using an SYBR Green I-based real-time PCR protocol. (**A**) Melting curve of RNA1; (**B**) Standard curve of RNA1; (**C**) Melting curve of RNA3; (**D**) Standard curve of RNA3.

**Figure 4 viruses-14-02818-f004:**
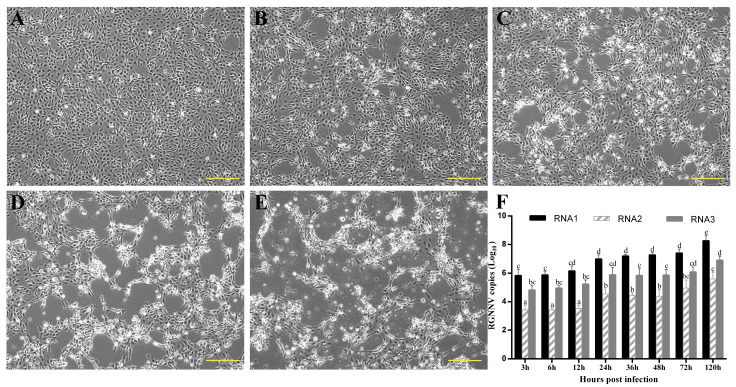
The changes in SSN-1 cell lesions and viral genome expression at different time points after NNV infection. (**A**–**E**): SSN-1 cell infected for 0 h, 24 h, 48 h, 72 h, 120 h. Scale bar = 100 μm. (**F**): qRT-PCR technique to detect intracellular viral genomes. Lowercase letters indicate significant differences between treatments (*p* < 0.05).

**Figure 5 viruses-14-02818-f005:**
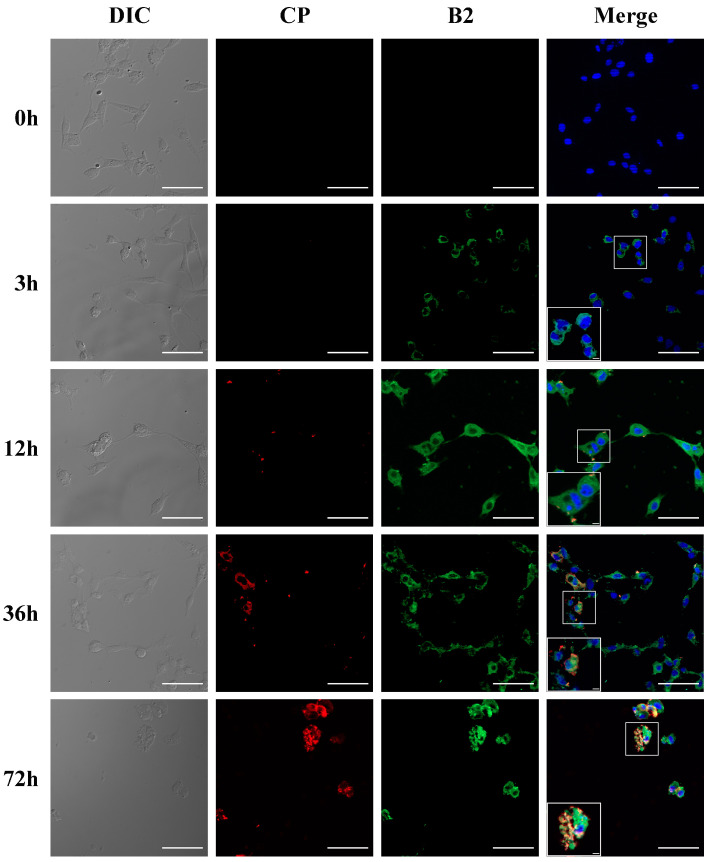
Indirect immunofluorescence analysis of Cp and B2 protein proliferation at different time points after NNV infection with SSN-1 Cells. Scale bars = 50 µm.

**Figure 6 viruses-14-02818-f006:**
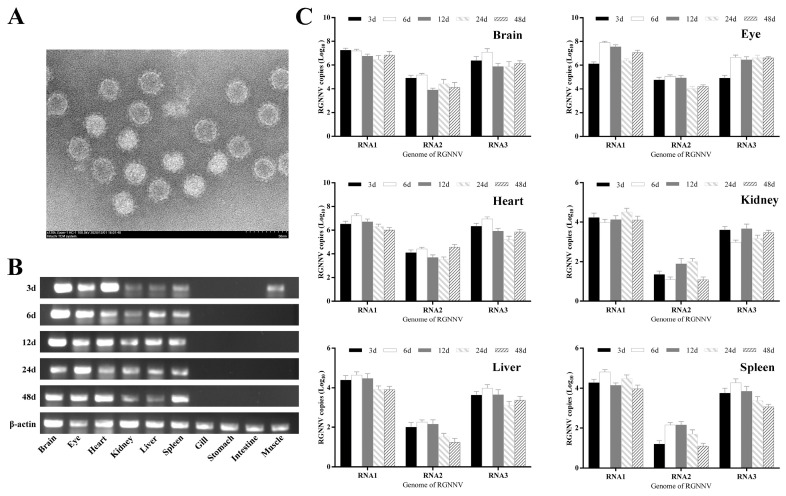
The changes in viral genome expression in tissues and organs of grouper larvae at different time points after NNV infection. (**A**) Electron micrograph of negatively stained virus particles pelleted from infected SSN-1 cells culture fluid which injected grouper larvae. (**B**) RT-PCR results of B2 protein gene expression in tissues of virus-infected grouper larvae; (**C**) qRT-PCR technique to detect changes in RNA1, RNA2 and RNA3 genome expression contents in tissues of virus-infected larvae.

**Figure 7 viruses-14-02818-f007:**
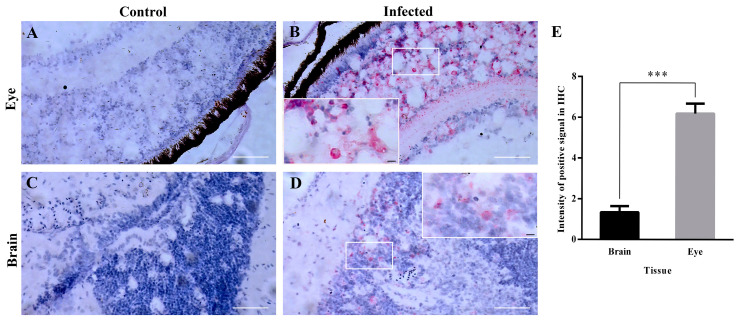
IHC detects and characterizes B2 protein expression in eye and brain tissues of grouper 3 d post-infection. (**A**) Healthy eye tissue; (**B**) NNV-infected eye tissue; (**C**) Healthy brain tissue; (**D**) NNV-infected brain tissue. Scale bar = 50 μm. (**E**) The intensity of positive signals in eye and brain. The data were processed by image J 1.51k. *** *p* < 0.001.

**Figure 8 viruses-14-02818-f008:**
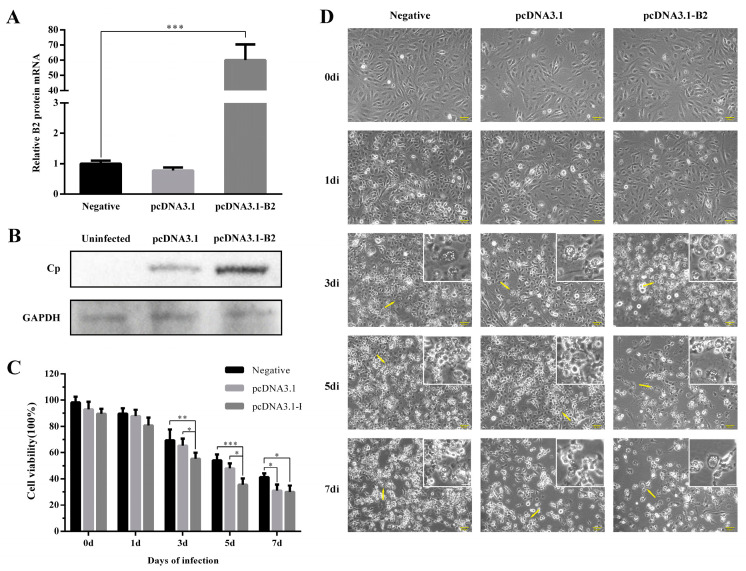
Overexpression of B2 protein on GF-1 cells following NNV infection. (**A**) The transfection efficiency of pcDNA3.1-B2 in GF-1 cells. *** *p* < 0.001. (**B**) Overexpression of B2 protein enhanced the expression of NNV Cp. Cells treated as above were harvested and lyzed for Western blot to validate expression changes of Cp. GAPDH served as the internal control and the uninfected group was designed as the control group. (**C**) Effect of B2 protein on cell viability. The detection followed the instruction of the CCK-8 kit. The data were shown from three biological replicates. The results were expressed as the percentage of viable cells and shown as means ± SD. *** *p* < 0.001, ** *p* < 0.01, * *p* < 0.05, the columns unlabeled presented not significant. (**D**) Characterization of CPE for continuous 7 days after infection with different transfection conditions in GF-1 cells. The magnified areas were indicated by the yellow arrows. Scale bar = 50 μm.

**Table 1 viruses-14-02818-t001:** Primers used in the present study.

Names	Sequences (5’ to 3’)	GenBank Number	Application
q-RNA1-F	GTATGTCGAGGAGCAACAGACC	MT157513.1	qRT-PCR
q-RNA1-R	GTTGAAATGTTCCTGGGGTAA		qRT-PCR
q-RNA3-F	TGACCGCGATCCAGGTAAACG	MT157513.1	qRT-PCR
q-RNA3-R	AAGAGGCGGGAACCGGCGTGAC		qRT-PCR
β-actin-F	CCA GAG CAA GAG GGG TAT C	KU200949.2	qRT-PCR
β-actin-R	GCT GTG GTG GTG AAG GAG T		qRT-PCR
NNV-B2-F	ATGGAACAAATCCAACAAGCGAT	MT157513.1	Plasmid Construction
NNV-B2-R	GTCCGTCTCCATCGGTTCCT		Plasmid Construction

## Data Availability

Not applicable.

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
