# Peer review of "Characterization of Nervous Necrosis Virus (NNV) Nonstructural Protein B2 and Its Enhancement on Virus Proliferation"

_viruses, 2022, doi:10.3390/v14122818_

Round 1
Reviewer 1 Report
In this MS, authors analyzed the effect of the nonstructural protein B2 encoded by the sub-genomic RNA3 in NNV on virus amplification. Results suggested that B2 protein acts as an early expressed protein during virus replication and proliferation and is involved in the early infection of NNV. This study is meaningful, but some doubts still need to be resolved.
1. line 13, in vivo and in vitro need to be italic.
2. Line 35, “Nodaviridae” should not be italicized.
3. In results 3.5, authors mentioned that B2 protein had a high expression level in brain, eye and heart and the genomes expression level in heart were as high as in eye and brain, which was novel to explore in the future. In 3.6, why did not detect the expression level of B2 protein in heart tissue by immunohistochemical analysis?
4. Line 387, the sentence of “the transfection of the eukaryotic plasmid pcDNA3.1-B2 in GF-1 cells was explored” was confused.
5. In Fig. 8C, what does the author think about the fact that there was no significant difference in cell survival between the pcDNA3.1-B2 and pcDNA3.1 transfected groups on day 7?
6. n 2.3, why use SSN-1 cells instead of GF-1 cells to study the dynamics of genome expression upon NNV infection? After all, the NNV mainly infects kinds of fish. The same question applies to the co-expression of B2 protein and Cp.
7. In Fig. 7, the expression level of B2 protein in the eyes and brain of NNV-infected grouper larvae was shown by immunohistochemistry analysis. Authors pointed out that the red positive signal was evident in the inner retinal choroidal cells and weaker in the brain grey matter. The author can add a bar chart to quantify this data.
8. In Fig. 8, authors detected the expression of NNV Cp to show B2 protein have a significant enhancement on virus proliferation. However, the transfection efficiency of pcDNA3.1-B2 in GF-1 cells was unknown. WB should be used to confirm the expression level of B2 protein after transfection in GF-1 cells.
Author Response
Sincerely thanks for your recommendations and suggestions about our manuscript “Characterization of nervous necrosis virus (NNV) nonstructural protein B2 and its enhancement on virus proliferation” (viruses- 2077741). We have revised the MS according to the suggestions. The revised parts are marked by “track changes” in the revised MS (viruses-2077741-R1). Detailed revisions are listed as follows:
Point 1: line 13, in vivo and in vitro need to be italic.
Response 1: Thank you for your reminder. The words has been modified (line 13) .
Point 2: Line 35, “Nodaviridae” should not be italicized.
Response 2: Thank you for your reminder. The words has been modified (line 35, 36) .
Point 3: In results 3.5, authors mentioned that B2 protein had a high expression level in brain, eye and heart and the genomes expression level in heart were as high as in eye and brain, which was novel to explore in the future. In 3.6, why did not detect the expression level of B2 protein in heart tissue by immunohistochemical analysis?
Response 3: NNV caused extensive necrosis in central nervous system. Eye and brain are the representative target organs in several researches. Therefore, we sampled this two organs for immunohistochemical analysis. In our future study, we will sample eye, brain and heart for the deeper exploration.
Point 4: Line 387, the sentence of “the transfection of the eukaryotic plasmid pcDNA3.1-B2 in GF-1 cells was explored” was confused.
Response 4: Thank you for your reminder. The sentence has been modified (line 387) .
Point 5: In Fig. 8C, what does the author think about the fact that there was no significant difference in cell survival between the pcDNA3.1-B2 and pcDNA3.1 transfected groups on day 7?
Response 5: As the results explored, B2 protein acted as an early expressed protein in the replication of virus. We detected the cell viability of GF-1 cells from 0 d to 7 d and discovered the significant differences between the group of pcDNA3.1 and pcDNA3.1-B2 on 3 di. It consistented with the results that B2 protein could advanced the death of cells in the early stage in Fig 4 and Fig 5. And from the expression of 72 h in Fig 5, it displayed that capsid protein began to plays the key role in cell death. The fish cell lines had low transfection efficiency, the impact of B2 protein might low in the late stage.
Point 6: In 2.3, why use SSN-1 cells instead of GF-1 cells to study the dynamics of genome expression upon NNV infection? After all, the NNV mainly infects kinds of fish. The same question applies to the co-expression of B2 protein and Cp.
Response 6 : Compared to GF-1 cells, SSN-1 showed typical CPE with cell rounding, grape-like clusters, detachment and meshwork after infection. It had more representation to study the dynamics on virus proliferation. And we used GF-1 cells only for the transfection of eukaryotic plasmid, as SSN-1 cells owned poor wall adhesion to support the experiment of overexpression.
Point 7: In Fig. 7, the expression level of B2 protein in the eyes and brain of NNV-infected grouper larvae was shown by immunohistochemistry analysis. Authors pointed out that the red positive signal was evident in the inner retinal choroidal cells and weaker in the brain grey matter. The author can add a bar chart to quantify this data.
Response 7: The new picture with quantified intensity of positive signal in Fig 7 was re-upload and relative information was added (line 244, 384-385).
Point 8: In Fig. 8, authors detected the expression of NNV Cp to show B2 protein have a significant enhancement on virus proliferation. However, the transfection efficiency of pcDNA3.1-B2 in GF-1 cells was unknown. WB should be used to confirm the expression level of B2 protein after transfection in GF-1 cells.
Response 8: Thank you for your reminder. For the failure to obtain the B2 protein from lysed cells, the transfection effciency was examined by qRT-PCR (Fig. 8A). And other relative information was added and modified in section 2.5 and section 3.7.
Reviewer 2 Report
Nervous necrosis virus (NNV) is responsible for causing diseases from larvae to adult marine and freshwater fish species throughout the world. In their manuscript, Zhang et al. characterized its B2 protein and found its enhancement on virus proliferation, which would increase the knowledge about the infection process, and find new drug/vaccination strategies against the infections. In this view, this manuscript could have the potential for the Viruses readers, however, it needs some further tests to confirm the findings.
Major points:
1. In Section 3.7, there was no data provided to confirm that the B2 overexpression was achieved in the cells after being transfected for 24 h.
2. In section 2.2, it mentioned that the cells were harvested and washed twice with PBS after 48 hpi for performing the western blot to detect the expression of NNV Cp. But according to the results in 3.7, GF-1 cells began to display CPE from the 3rd day after infection. Why was this time point (48 h) chosen? How about the virus loads for the other time points? The cells for all of the timepoint should be harvested for western blot to validate expression changes of Cp.
3. Fig 8A, the un-transfected cells (negative control) showed a comparable degree of CPE to the other two groups. But the Fig 8B WB result showed no detected specific bands for the negative control cells, it needs explanation.
4. In the discussion section, the author mentioned that the B2 protein was hard to obtain from lysed cells after infection due to its small molecular (8.5 kDa). Have you attempted to detect the B2 protein using flow cytometry with the antibody you obtained? Here is the reference that may help you. [ McSharry JJ. Analysis of virus-infected cells by flow cytometry. Methods. 2000 Jul;21(3):249-57. doi: 10.1006/meth.2000.1005. PMID: 10873479.]
Minor points:
1. Some units in this manuscript were not written in the standard way, such as “105 TCID50/ml” in line 122 should be “TCID50/mL”. And there should be a space between the previous number and the unit, such as line 340/344/346, 3h/12h/72h//12h should be 3 h/12 h/72 h/12 h. Please check and modify the whole manuscript.
2. Line 126-128, “The supernatant was taken from the remaining slides and the adherent cells were washed once with DEPC for qRT-PCR of RNA1, RNA2 and RNA3 in a closed water system.” should be “The supernatant was taken from the remaining slides and the adherent cells were washed once with diethylpyrocarbonate (DEPC)-treated water for qRT-PCR of RNA1, RNA2, and RNA3 in a closed water system.” Therefore, it just needs to keep the abbreviation (DEPC) for the following test in line 148.
3. Resolution of Figure 6 is too low.
Author Response
Sincerely thanks for your recommendations and suggestions about our manuscript “Characterization of nervous necrosis virus (NNV) nonstructural protein B2 and its enhancement on virus proliferation” (viruses- 2077741). We have revised the MS according to the suggestions. The revised parts are marked by “track changes” in the revised MS (viruses-2077741-R1). Detailed revisions are listed as follows:
Point 1: In Section 3.7, there was no data provided to confirm that the B2 overexpression was achieved in the cells after being transfected for 24 h.
Response 1: For the failure to obtain the B2 protein from lysed cells, the transfection effciency was examined by qRT-PCR (Fig. 8A). And other relative information was added and modified in section 2.5 and section 3.7.
Point 2: In section 2.2, it mentioned that the cells were harvested and washed twice with PBS after 48 hpi for performing the western blot to detect the expression of NNV Cp. But according to the results in 3.7, GF-1 cells began to display CPE from the 3rd day after infection. Why was this time point (48 h) chosen? How about the virus loads for the other time points? The cells for all of the timepoint should be harvested for western blot to validate expression changes of Cp.
Response 2: The results of CPE was presented to show the differences between groups. It displayed the function of B2 protein on the early extensive death of cells. We found that GF-1 cells began to display CPE on day 2, but there had little difference between groups to display. And on day 3, the significant difference on CPE results showed. So we chose the time point of CPE began to display (48 h) for western blot to explore the function of B2 protein and chose the other time points to show the differences. And as the early expressed nonstructual peotein, we designed to investigate its function in the early stage on virus replication.
Point 3: Fig 8A, the un-transfected cells (negative control) showed a comparable degree of CPE to the other two groups. But the Fig 8B WB result showed no detected specific bands for the negative control cells, it needs explanation.
Response 3: Sorry for the mislead the comprehension. We had modified the Figure 8 and sentences in the manuscript (line 416).
Point 4: In the discussion section, the author mentioned that the B2 protein was hard to obtain from lysed cells after infection due to its small molecular (8.5 kDa). Have you attempted to detect the B2 protein using flow cytometry with the antibody you obtained? Here is the reference that may help you. [ McSharry JJ. Analysis of virus-infected cells by flow cytometry. Methods. 2000 Jul;21(3):249-57. doi: 10.1006/meth.2000.1005. PMID: 10873479.]
Response 4: Thank you for your advice. We had attempt to recognize the capsid protein by monoclonal antibody using flow cytometry in our previously pre-experiment. Regretly the results were too poorly to submit in the paper.
Point 5: Some units in this manuscript were not written in the standard way, such as “105 TCID50/ml” in line 122 should be “TCID50/mL”. And there should be a space between the previous number and the unit, such as line 340/344/346, 3h/12h/72h//12h should be 3 h/12 h/72 h/12 h. Please check and modify the whole manuscript.
Response 5: Thank you for reminder. The format was modified in whole manuscript.
Point 6: Line 126-128, “The supernatant was taken from the remaining slides and the adherent cells were washed once with DEPC for qRT-PCR of RNA1, RNA2 and RNA3 in a closed water system.” should be “The supernatant was taken from the remaining slides and the adherent cells were washed once with diethylpyrocarbonate (DEPC)-treated water for qRT-PCR of RNA1, RNA2, and RNA3 in a closed water system.” Therefore, it just needs to keep the abbreviation (DEPC) for the following test in line 148.
Response 6: Thank you for reminder. The sentences were modified (line 127, 148).
Point 7: Resolution of Figure 6 is too low.
Response 7: The new picture was re-upload.
Round 2
Reviewer 2 Report
It can be accepted.
Author Response
Thank you and the reviewers for giving us so valuable comments on the manuscript ID: viruses-2077741, entitled “Characterization of nervous necrosis virus (NNV) nonstructural protein B2 and its enhancement on virus proliferation”. These comments are all valuable and very helpful for revising and improving our paper. We made a careful revision in which we fully addressed the issues point by point raised from the reviewers’ comments. And we hope that the revised manuscript is acceptable for publication in “Viruses”.
All modifications were marked in “Track changes” mode in revised Manuscript. We deeply appreciate your consideration of our manuscript.
Thank you and best regards.
